# Learning Articulated Rigid Body Dynamics with Lagrangian Graph Neural Network

**Ravinder Bhattoo**
Department of Civil Engineering
Indian Institute of Technology Delhi
`cez177518@iitd.ac.in`

**Sayan Ranu**
Department of Computer Science and Engineering
Yardi School of Artificial Intelligence
Indian Institute of Technology Delhi
`sayanranu@iitd.ac.in`

**N. M. Anoop Krishnan**
Department of Civil Engineering
Yardi School of Artificial Intelligence
Indian Institute of Technology Delhi
`krishnan@iitd.ac.in`

## Abstract

Lagrangian and Hamiltonian neural networks (LNNs and HNNs, respectively) encode strong inductive biases that allow them to outperform other models of physical systems significantly. However, these models have, thus far, mostly been limited to simple systems such as pendulums and springs or a single rigid body such as a gyroscope or a rigid rotor. Here, we present a Lagrangian graph neural network (LGNN) that can learn the dynamics of articulated rigid bodies by exploiting their topology. We demonstrate the performance of LGNN by learning the dynamics of ropes, chains, and trusses with the bars modeled as rigid bodies. LGNN also exhibits generalizability—LGNN trained on chains with a few segments exhibits generalizability to simulate a chain with large number of links and arbitrary link length. We also show that the LGNN can simulate unseen hybrid systems including bars and chains, on which they have not been trained on. Specifically, we show that the LGNN can be used to model the dynamics of complex real-world structures such as the stability of tensegrity structures. Finally, we discuss the non-diagonal nature of the mass matrix and its ability to generalize in complex systems.

## 1 Introduction and Related Works

Movements of a robotic arm, rolling ball, or falling chain can be characterized by rigid body motion [1, 2]. Understanding the dynamics of the motion is crucial in several applications including robotics, human-robot interaction, planning, and computer graphics [3, 1]. Traditionally, the rigid body mechanics is studied in the framework of classical mechanics, which relies on either force-based or energy-based approaches [4]. Force-based approaches involve the computation of all the unknown forces based on the equations of equilibrium and hence is cumbersome for large structures. Energy-based approaches present an elegant formalism which involve the computation of a scalar quantity representing the state of a system, namely, Lagrangian ($\mathcal{L} = \mathcal{T} - \mathcal{V}$), which is the difference between the kinetic ($\mathcal{T}(q, \dot{q})$) and potential ($\mathcal{V}(q)$) energies, or Hamiltonian ($\mathcal{H} = \mathcal{T} + \mathcal{V}$), which represents the total energy of the system. This scalar quantity can, in turn, be used to predict the dynamics of the system. However, the functional form governing this scalar quantity may not be

---

The code is available at https://github.com/M3RG-IITD/rigid_body_dynamics_graph

36th Conference on Neural Information Processing Systems (NeurIPS 2022).

known *a priori* in many cases [5]. Thus, learning the dynamics of rigid bodies directly from the trajectory can simplify and accelerate the modeling of these systems [5, 6, 7, 8].

Learning the dynamics of particles has received much attention recently using physics-informed approaches [9]. Among these, Lagrangian neural networks (LNNs) and Hamiltonian neural networks (HNNs) are two physics-informed neural networks with strong inductive biases that outperform other learning paradigms of dynamical systems [10, 11, 12, 8, 6, 13, 7, 14]. In this approach, a neural network is trained to learn the $\mathcal{L}$ (or $\mathcal{H}$) of a system based on its configuration $(q, \dot{q})$. The $\mathcal{L}$ is then used along with the *Euler-Lagrange (EL)* equation to obtain the time evolution of the system. Note that the training of LNNs is performed by minimizing the error on the predicted trajectory with respect to the actual trajectory. Thus, LNNs can effectively learn the Lagrangian directly from the trajectory of a multi-particle system [6, 13].

Most of the works on LNN has focused on relatively simpler particle-based systems such as springs and pendulums [15, 16, 6, 13, 7, 10, 17]. This approach models a rigid body, for instance a ball, as a particle and predicts the dynamics. This approach thus ignores the additional rotational degrees of freedom of the body due to its finite volume. Specifically, while a particle in 3D has three degrees of freedom (translational), a rigid body in 3D has six degrees of freedom (translational and rotational). Thus, the dynamics and energetics associated with these degrees of motions are lost by modeling a rigid body as a particle. To the best of authors' knowledge, thus far, only one work has attempted to learn rigid body dynamics using LNNs and HNNs, where it was demonstrated the dynamics of simple rigid bodies such as a gyroscope or rotating rotor can be learned [13]. However, the LNNs used in this work, owing to their fully connected MLP architecture, are transductive in nature. An LNN trained on a double-pendulum system or 3-spring system can be used only for the same system and does not generalize to a different system size such as 3-pendulum or 5-spring, respectively. In realistic situations the number of particles in a system can vary arbitrarily, and accordingly, a large number of trained models might be required to model these systems.

An alternate approach to model these systems would be to use a graph neural network (GNN) [18, 19, 5, 15, 16], which, once trained, can generalize to arbitrary system sizes. GNNs have been widely used to model physical and atomic systems extensively due to their inductive bias [20, 21, 22, 15, 16]. GNNs have also been used to model rigid bodies mainly following two approaches, namely, *particle-based* [19] and *lumped mass* [22, 23] methods. In the first approach, a rigid body is discretized into finite number of particles and the motion of the individual particles are learned to predict the dynamics of rigid body [19]. Note that this approach is philosophically similar to mess-less methods such as smoothed-particle hydrodynamics (SPH) [24] or peridynamics (PD) [25], where the time-evolution of a continuum body is simulated by discretizing the domain using particles. This approach [19], although useful, have several limitations, namely, it does not (i) conserve physical quantities such as energy when simulated over a long duration, and (ii) generalize to a different timestep of forward simulation than the one on which it is trained. In the second approach, a rigid body is modeled as a lumped mass [22, 26], the dynamics of which is learned by assuming this lumped mass as a particle. For instance, the dynamics of a chain is modeled by discretizing the chain to smaller segments and modeling each segment as a lumped mass. As mentioned earlier, this approach leads to the loss of additional degrees of freedom that are associated with a rigid body.

Here, we present a Lagrangian graph neural network (LGNN) framework that can learn the dynamics of rigid bodies. Specifically, exploiting the topology of a physical system, we show that a rigid body can be modeled as a graph. Further, the Lagrangian of the graph structure can be learned directly by minimizing the loss on the predicted trajectory with respect to the actual trajectory of the system. The major contributions of the work are as follows.

- **Topology aware modeling of rigid body.** We present a graph-based model for articulated rigid bodies such as in-extensible ropes, chains, or trusses. Further, we demonstrate using LGNN that the dynamics of these systems can be learned in the Lagrangian framework.
- **Generalizability to arbitrary system sizes.** We show that LGNN can generalize to arbitrary system sizes once trained.
- **Generalizability to complex unseen topology.** We demonstrate that the LGNN can generalize to unseen topology, that is, links with varying lengths, a combination of truss and chain structures, and different boundary conditions.

Altogether, we demonstrate that LGNN can be a strong framework for simulating the dynamics of articulated rigid bodies.

## 2 Dynamics of Rigid Bodies

The dynamics of a physical system can be represented as $\ddot{q} = F(q, \dot{q}, t)$, where $q, \dot{q} \in \mathbb{R}^D$ is a function of time ($t$) for a system with $D$ degrees of freedom. The future states or *trajectory* of the system can be predicted by integrating these equations to obtain $q(t+1)$ and so on. While there are several physics-based methods for generating the *dynamics* of the system such as d'Alembert's principle, Newtonian, Lagrangian, or Hamiltonian approaches, all these approaches result in the equivalent sets of equations [3].

The two broad paradigms for modeling the dynamics involve force- and energy-based approaches. Energy-based approaches is an elegant framework, which relies on the computation of a single scalar quantity, for instance energy, that represents the state of system. The dynamics of the system is, in turn, computed based on this scalar quantity. Among the energy-based approaches, Lagrangian formulation has been widely used to predict the dynamics of particles and rigid bodies by computing the Lagrangian $\mathcal{L}$ of the system. The standard form of Lagrange's equation for a system with *holonomic* constraints is given by $\frac{d}{dt}\left(\frac{\partial \mathcal{L}}{\partial \dot{q}}\right) - \left(\frac{\partial \mathcal{L}}{\partial q}\right) = 0$, and the Lagrangian is $\mathcal{L}(q, \dot{q}, t) = \mathcal{T}(q, \dot{q}, t) - \mathcal{V}(q, t)$ with $\mathcal{T}(q, \dot{q}, t)$ and $\mathcal{V}(q, t)$ representing the total kinetic energy of the system and the potential function from which generalized forces can be derived. Accordingly, the dynamics of the system can be represented using EL equations as $\ddot{q}_i = \left(\frac{\partial^2 \mathcal{L}}{\partial \dot{q}_i^2}\right)^{-1}\left[\frac{\partial \mathcal{L}}{\partial q_i} - \left(\frac{\partial \mathcal{L}}{\partial \dot{q}_i \partial q_i}\right)\dot{q}_i\right]$.

**Modified Euler-Lagrange Equation.** A modified version of the EL can be used in cases where some of the terms involved in the equation can be decoupled. This formulation allows explicit incorporation of constraints (holonomic and Pfaffian) and additional dissipative terms for friction or drag [3, 1]. In rigid body motion, Pfaffian constraints can be crucial in applications such as multi-fingered grasping where, the velocity of two or more fingers are constrained so that the combined geometry formed is able to catch or hold an object. A generic expression of constraints for these systems that accounts for both holonomic and Pfaffian can be $A(q)\dot{q} = 0$, where, $A(q) \in \mathbb{R}^{k \times D}$ represents $k$ velocity constraints. In addition, drag, friction or other dissipative terms of a system can be expressed as an additional forcing term in the EL equation. It is worth noting that EL equation, by nature, is energy conserving. Hence, the additional dissipative terms are crucial for modeling realistic systems with friction and drag. If these terms are not included, the system will essentially try to simulate an energy preserving trajectory, thereby resulting in huge errors in the dynamics [17].

Considering the additional forces mentioned above, the modified EL equation can be written as:

$$\frac{d}{dt}\nabla_{\dot{q}}\mathcal{L} - \nabla_q\mathcal{L} + A^T(q)\lambda - \Upsilon - F = 0 \tag{1}$$

where $A^T$ forms a non-normalized basis for the constraint forces, $\lambda \in \mathbb{R}^k$, known as the Lagrange multipliers, gives the relative magnitudes of these force constraints, $\Upsilon$ represents the non-conservative forces, such as friction or drag, which are not directly derivable from a potential, and $F$ represents any external forces acting on the system. This equation can be modified to obtain $\ddot{q}$ as:

$$\ddot{q} = M^{-1}\left(-C\dot{q} + \Pi + \Upsilon - A^T(q)\lambda + F\right) \tag{2}$$

where $M = \frac{\partial}{\partial \dot{q}}\frac{\partial \mathcal{L}}{\partial \dot{q}}$ represents the mass matrix, $C = \frac{\partial}{\partial q}\frac{\partial \mathcal{L}}{\partial \dot{q}}$ represents Coriolis-like forces, and $\Pi = \frac{\partial \mathcal{L}}{\partial q}$ represents the conservative forces derivable from a potential. Differentiating the constraint equation gives $A(q)\ddot{q} + \dot{A}(q)\dot{q} = 0$. Solving $\lambda$ (see A.2) and substituting in Eq. 2, we obtain $\ddot{q}$ as

$$\ddot{q} = M^{-1}\left(\Pi - C\dot{q} + \Upsilon - A^T(AM^{-1}A^T)^{-1}\left(AM^{-1}(\Pi - C\dot{q} + \Upsilon + F) + \dot{A}\dot{q}\right) + F\right) \tag{3}$$

For a system subjected to these forces, the dynamics can be learned using LNN by minimizing the loss on the predicted and observed trajectory, where the predicted acceleration $\hat{\ddot{q}}$ is obtained using the Equation 3. It is worth noting that in this equation, $M, C$, and $\Pi$ can be directly derived from the $\mathcal{L}$. Constraints on the systems are generally known as they generally form part of the topology. It is worth noting that there are some recent works that focus on learning constraints as well [8].

# 3 Lagrangian Mechanics for Articulated Rigid Bodies

In the case of particle systems such as spring or pendulum systems, the approach mentioned in Sec.2 can be directly used in conjunction with an LNN to learn the dynamics. In this case, the mass matrix $M(q)$ remains constant with only diagonal entries $m_{ii}$ in Cartesian coordinates. Inducing this as a prior knowledge, wherein the masses are parameterized as a diagonal matrix is shown to simplify the learning process [13]. However, in the case of an articulated rigid body, the mass matrix is non-diagonal in the Cartesian coordinates. Further, the kinetic energy term $\mathcal{T}$ becomes a function of both position and velocity. In other words, the kinetic energy also becomes a function of the topology. This makes learning the dynamics a complex problem especially in real-world complex structures such as trusses or tensegrities, which are a combination of bars, ropes, and chains.

To this extent, we briefly review the mechanics of a falling rope or chain as an example. Note that simple rigid bodies such as a gyroscope or rotating rotor has already been studied using LNNs [13]. Of our special interest are articulated rigid bodies that can be arbitrarily large such as chains, ropes or trusses, that can be divided into smaller constituent members. This is because, it is generally assumed that extending LNNs to large structures is a challenging problem [17]. Traditionally, the mechanics of chains or ropes are modeled using discrete models [2]. Figure 1 shows a discrete model of a rope of mass $M$ and length $L$. The rope is discretized into $n$ cylindrical rods or segments each having a mass $m_i = M/n$ and length $l_i = L/n$. These segments are considered to be rigid, and with a finite uniform cross-sectional area and volume. In order to replicate realistic dynamics of a rope, the $l_i$ should be significantly smaller than $L$. Note that in the case of a chain or truss, such artificial discretization is not required and the bars associated with each segment can be directly considered as a rigid body.

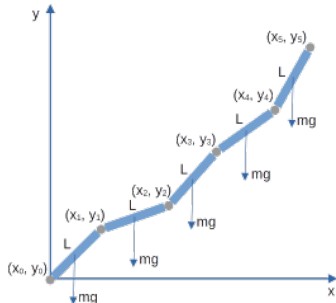

Figure 1: Chain considering rigid body dynamics. Spheres represent the connections and bars represent the links of the chain.

To formulate the $\mathcal{L}$, the generalized coordinates with orientation of each link represented by $\phi_i = tan^{-1}\left(\frac{y_i - y_{i-1}}{x_i - x_{i-1}}\right)$ can be considered. Placing the origin at the beginning of first segment (see Figure 1), the center of mass of $i^{th}$ segment $(x_i^{cm}, y_i^{cm})$ can be written in terms of generalized coordinates as

$$x_i^{cm} = \sum_{j=1}^{i-1} l_j \cos\phi_j + \frac{1}{2}l_i \cos\phi_i, \qquad y_i^{cm} = \sum_{j=1}^{i-1} l_j \sin\phi_j + \frac{1}{2}l_i \sin\phi_i \qquad (4)$$

Accordingly, the kinetic energy of the system is given by [2]

$$\mathcal{T} = \frac{1}{2}\sum_{i=1}^{n}\left(m_i(\dot{x}_{i,cm}^2 + \dot{y}_{i,cm}^2) + I_i\dot{\phi}_i^2\right) \qquad (5)$$

where $I_i = \frac{1}{12}m_i l_i^2$ represents the moment of inertia of the rigid segment $i$. Similarly, the potential energy of the system can be expressed as:

$$\mathcal{V} = \sum_{i=1}^{n} m_i g y_i^{cm} \qquad (6)$$

where $g$ represents the acceleration due to gravity. Finally, the Lagrangian of the system can be obtained as $\mathcal{L} = \mathcal{T} - \mathcal{V}$, which can be substituted in the EL equation to obtain the dynamics of the rigid body.

To learn the dynamics of an articulated rigid body, we employ the approach shown in Figure 2. Specifically, we model a physical system as a graph. Further, the Lagrangian of system is learned by decoupling the potential and kinetic energy, each of which are learned by two GNNs, namely, $\mathcal{G}_\mathcal{V}$ and $\mathcal{G}_\mathcal{T}$. Finally, the Lagrangian is computed as $\mathcal{L} = \mathcal{T} - \mathcal{V}$. This framework is trained end-to-end based by minimizing the loss on the acceleration predicted by the LGNN using EL equation with respect to the ground truth. In this section, we describe the LGNN architecture for rigid bodies in detail (See

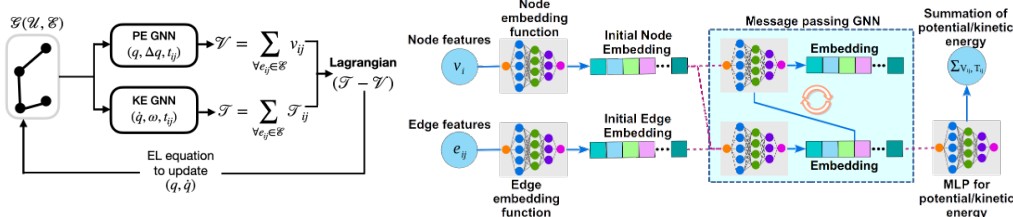

Figure 2: (a) Learning articulated rigid body dynamics with LGNN (b) Architecture of LGNN for rigid bodies.

Figure 2 for an overview). We empirically show that the dynamics of a rigid body can be learned by LGNN. In addition, due to the inductive nature of the graph architecture, once trained on a small system, LGNN can generalize to arbitrary system sizes and topology.

**Graph structure.** Figure 1 shows a chain. The (undirected) graph of the physical system is constructed by considering the bars/segments of the chain as the edges and the connections as nodes. Here, edges represent the rigid bodies and nodes represent the connection between these rigid bodies. This is in contrast to earlier approaches used for particle-based systems, where node represented the particle position and edge represented the connections between them. Hereon, we use the notation $\mathcal{G}(\mathcal{U}, \mathcal{E})$ to to represent the graph representation of a rigid body with $\mathcal{U}$ and $\mathcal{E}$ as its node and edge sets.

**Overview of the architecture.** As shown in Figure 2, we use two GNNs; one to predict the potential energies and the other to predict kinetic energies. From these predictions the Lagrangian is computed. The error on the Lagrangian is minimized through an RMSE loss function to jointly train both the GNNs. The architecture of both the GNNs, shown in Figure 2, are identical. Note that the specific graph architecture used in the present work is inspired from previous works on LGNNs for particle-based systems [15, 16].

**Input features.** Each node $u_i \in \mathcal{U}$ is characterized by its position $q_i = (x_i, y_i, z_i)$, and velocity ($\dot{q}_i$). Each edge $e_{ij}$ is characterized by its *type* $t_{ij}$, and the relative differences in the *positions* ($\Delta q_{ij} = q_i - q_j$) of its connecting nodes, and $\omega_{ij} = \Delta q_{ij} \times \Delta \dot{q}_{ij}$. The type $t_{ij}$ is a discrete variable and is useful in distinguishing edges of different characteristics within a system (Ex. moment inertia or area of cross section of the edge). Note that the velocity of a rigid body represented by an edge is a function of the velocities of its end points in two and three dimensional spaces. Hence, we do not explicitly track edge velocities.

**Pre-Processing.** In the pre-processing layer, we construct a dense vector representation for each node $v_i \in \mathcal{U}$ and edge $e_{ij} \in \mathcal{E}$ using `MLP`s (multi-layer perceptrons). The exact operation for potential energy is provided below in Eqs.7-8. For kinetic energy, we input $\dot{q}_i$ in Eq 7 instead of $q_i$ and $\omega_{ij}$ in Eq. 8 instead of $\Delta q_{ij}$.

$$\mathbf{h}_i^0 = \texttt{squareplus}(\texttt{MLP}(q_i)) \qquad (7)$$

$$\mathbf{h}_{ij}^0 = \texttt{squareplus}(\texttt{MLP}(\texttt{one-hot}(t_i), \Delta q_{ij})) \qquad (8)$$

`squareplus` is an activation function.

**Message passing.** To infuse structural information in the edge and node embeddings, we perform $L$ layers of message passing, wherein the embedding in each layer $l \in [1, \cdot, L]$ is computed as follows:

$$\mathbf{h}_{ij}^{l+1} = \texttt{squareplus}\left(\texttt{MLP}\left(\mathbf{h}_{ij}^l + \mathbf{W}_{\mathcal{E}}^l \cdot \left(\mathbf{h}_i^l || \mathbf{h}_j^l\right)\right)\right) \qquad (9)$$

Here, $\mathbf{W}_{\mathcal{E}}^l$ is a layer-specific learnable weight vector and $\|$ represents concatenation operation. The node embeddings in a given layer $l$ are learned as follows:

$$\mathbf{h}_i^{l+1} = \texttt{squareplus}\left(\texttt{MLP}\left(\mathbf{h}_i^l + \sum_{j \in \mathcal{N}_i} \mathbf{W}_{\mathcal{U}}^l \cdot \mathbf{h}_{ij}^l\right)\right) \qquad (10)$$

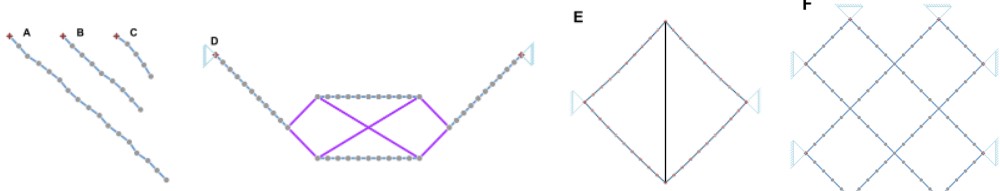

Figure 3: (Topologies of the $n$-segment chain/rope structures with $n =$ (a) 4, (b) 8, and (c) 16. Topologies of complex tensegrity structures (d) T1, (e) T2, and (f) T3 made of chains and rods. Note that the 4-link system in (a) is used train the LGNN and the trained model is used for inferring the dynamics on all other systems.

Here, $\mathcal{N}_i = \{u_j \mid (u_i, u_j) \in \mathcal{E}\}$ denotes the edges incident on node $u_i$. Similar to $\mathbf{W}_{\mathcal{E}}^l$, $\mathbf{W}_{\mathcal{U}}^l$ is a layer-specific learnable weight vector, which performs a linear transformation on the embedding of each incident edge. Following $L$ layers of message passing, the final node and edge representations in the $L^{th}$ layer are denoted by $\mathbf{z}_i = \mathbf{h}_i^L$ and $\mathbf{z}_{ij} = \mathbf{h}_{ij}^L$ respectively.

**Potential and kinetic energy prediction.** The predicted potential energy of each edge (rigid body) is computed by passing its final layer embedding through an MLP, i.e., $\widehat{v_{ij}} = \texttt{MLP}(\mathbf{z}_{i,j})$. The global predicted potential energy of the rigid body system is therefore the sum of the individual energies, i.e., $\widehat{\mathcal{V}} = \sum_{\forall e_{ij} \in \mathcal{E}} \widehat{v}_{ij}$. For kinetic energy, the computation is identical except that it occurs in the other GNN with parameters optimized for kinetic energy.

**Loss function.** The predicted Lagrangian is simply the difference between the predicted kinetic energy and the potential energy. Using Euler-Lagrange equations, we obtain the predicted acceleration $\widehat{\ddot{q}}_i(t)$ for each node $u_i$. The ground truth acceleration is computed directly from the ground truth trajectory using the Verlet algorithm as:

$$\ddot{q}_i(t) = \frac{1}{(\Delta t)^2}[q_i(t + \Delta t) + q_i(t - \Delta t) - 2q_i(t)] \tag{11}$$

The parameters of the GNNs are trained to minimize the RMSE loss over the entire trajectory $\mathbb{T}$:

$$\mathbb{L} = \frac{1}{|\mathcal{U}|} \left( \sum_{\forall u_i \in \mathcal{U}} \sum_{t=2}^{|\mathbb{T}|} \left( \ddot{q}_i(t) - \left( \widehat{\ddot{q}}_i(t) \right) \right)^2 \right) \tag{12}$$

Since the integration of the equations of motion for the predicted trajectory is also performed using the same algorithm as: $q(t + \Delta t) = 2q(t) - q(t - \Delta t) + \ddot{q}(\Delta t)^2$, this method is equivalent to training from trajectory/positions.

## 4 Empirical Evaluation

In this section, we evaluate the ability of LGNN to learn rigid body dynamics. In addition, we evaluate the ability of LGNN to generalize to larger unseen system sizes, complex topology, and realistic structures such as tensegrity.

### 4.1 Experimental setup

• **Simulation environment.** All the training and forward simulations are carried out in the JAX environment [21]. The graph architecture is implemented using the jraph package [27]. All the codes related to dataset generation and training are available in https://github.com/M3RG-IITD/rigid_body_dynamics_graph.
**Software packages:** numpy-1.20.3, jax-0.2.24, jax-md-0.1.20, jaxlib-0.1.73, jraph-0.0.1.dev0
**Hardware:** Memory: 16GiB System memory, Processor: Intel(R) Core(TM) i7-10750H CPU @ 2.60GHz

•**Baselines.** As outlined earlier, there are very few works on rigid body simulations using graph-based approaches, where the graph is used to model the topology of the rigid body. To compare the

performance of LGNN, we employ three baselines, namely, (i) a graph network simulator GNS, (ii) a Lagrangian graph network (LGN), and (iii) constrained Lagrangian neural network (CLNN). GNS employs a full graph network architecture [5, 12, 19] to predict the update in the position and velocity of node based on the present position and velocity. GNS has been shown to be a versatile model with the capability to simulate a wide range of physical systems [19]. LGN and CLNN employs the exact same equations as LGNN for computing the acceleration and trajectory and hence has the same inductive biases as LGNN in terms of the training and inference. However, while LGN employs a full graph network, CLNN employs a feed-forward multilayer perceptron. Details of the architectures and the hyperparameters of the baselines are provided in the Appendix A.5 and Appendix A.6, respectively.

• **Datasets and systems.** To evaluate the performance LGNN, we selected $n$-chain/rope systems, where $n = (4, 8, 16)$. All the graph based models are trained only on 4-segment chain system, which are then evaluated on other system sizes. Further, to evaluate the zero-shot generalizability of LGNN to large-scale unseen systems, we simulate 8-, and 16-segment chain systems. Further, to push the limits of LGNN, we evaluate the model trained on 4-segment chain on a 100-link system, and to complex shaped topologies involving truss members (long rigid members) and chains (short rigid members), which have more than 40 segments (see Figure 3). The mass $m_i$ and moment of inertia $I_i$ of all the members are maintained to be the same for all the segments irrespective of their length. To evaluate the generalizability to realistic systems, we also evaluate the performance on a 4-link system with different link properties and also with an external drag. The details of the experimental systems are given in Appendix A.1. Further, the detailed data-generation procedure is given in the Appendix A.4.

• **Evaluation Metric.** Following the work of [13], we evaluate performance by computing the relative error in **(1)** the trajectory, known as the *rollout error*, given by $RE(t) = ||\hat{q}(t) - q(t)||_2/(||\hat{q}(t)||_2 + ||q(t)||_2)$ and **(2)** *energy violation error* given by $||\hat{\mathcal{H}} - \mathcal{H}||_2/(||\hat{\mathcal{H}}||_2 + ||\mathcal{H}||_2)$. In addition, we also compute the geometric mean of rollout and energy error to compare the performance of different models [13]. Note that all the variables with a hat, for example $\hat{x}$, represent the predicted values based on the trained model and the variables without hat, that is $x$, represent the ground truth.

• **Model architecture and training setup.** For the graph architectures, namely, LGNN and GNS, all the neural networks are modeled as one hidden layer MLPs with varying number of hidden units. For all the MLPs, a square-plus activation function is used due to its double differentiability. In contrast to the earlier approaches, here, the training is not performed on trajectories. Rather, it is performed on 10000 data points generated from 100 trajectories for all the models. This dataset is divided randomly in 75:25 ratio as training and validation set. The model performance is evaluated on a forward trajectory, a task it was not explicitly trained for, of $1s$. Note that this trajectory is ∼2-3 orders of magnitude larger than the training trajectories from which the training data has been sampled. The dynamics of $n$-body system is known to be chaotic for $n \geq 2$. Hence, all the results are averaged over trajectories generated from 100 different initial conditions. Detailed model architecture associated with each of the models and the hyperparameters used in the training are provided in the Appendices A.5 and A.6, respectively.

## 4.2 Comparison with baselines

**Model performance.** To compare the performance of LGNN with baselines, GNS, LGN [12, 6] and CLNN [13], we evaluate the evolution of energy violation and rollout error. It worth noting that GNS and LGN have been demonstrated only particle-based systems and not on rigid bodies. Hence, to make a fair comparison, we give the same node and edge input features as provided for the LGNN for both GNS and LGN, while training. All the models are trained on a 4-link system and evaluated on all other systems. In the case of CLNN, due to the fully connected architecture, the model is no inductive in nature. Hence, the model is trained and tested on the same system only, that is, the 4-link system. Detailed architecture of each of these systems are provided in Appendix A.5. Figure 4 shows the error in energy and rollout for LGNN, GNS, LGN, and CLNN. We observe that GNS, LGN and CLNN have a larger error in comparison to LGNN as shown in Figure 4 for both energy and rollout error, establishing the superiority of LGNN. To test the ability of LGNN to learn more complex systems, we consider two additional experiments. Specifically, two similar 4-link systems, one with varying masses and moment of inertia, and the other subjected to a linear drag are evaluated in the

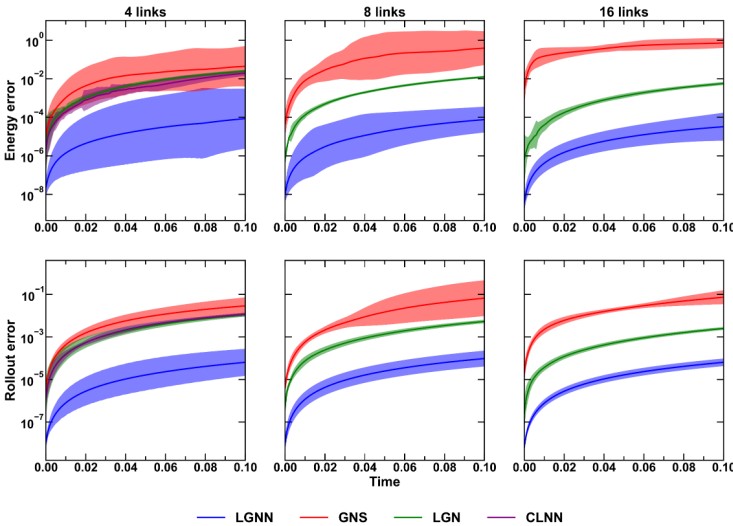

Figure 4: Energy and rollout error of $n-$link chains predicted using LGNN, GNS, LGN and CLNN in comparison with ground truth for $n = 4, 8, 16$. Note that LGNN, GNS and LGN is trained only on the 4-link chain and predicted on all the systems while CLNN was trained and predicted only on 4-link system. Shaded region shows the 95% confidence interval based on 100 forward simulations with random initial conditions.

Appendix A.7. Figures 8 and 14 show that LGNN is able to infer the dynamics in both these systems, respectively.

**Generalizability to different system sizes.** Now, we analyze the performance of LGNN, trained on 4-link segment, on 8- and 16-link segments. We observe that LGNN exhibits comparable performances with respect to the 4-segment model, in terms of both energy violation error and rollout error, on systems with 8-, and 16-segments that are unseen by the model. In contrast, GNS exhibits relatively increased error in energy violation error and rollout error, although the error in LGN remains comparable for all systems. This suggests that the inductive bias in terms of the EL equations prevent the accumulation of error and allow improved generalization. However, the error in LGN is still orders magnitude higher than LGNN. This suggests that the architecture employed in LGNN is leading improved learning of the dynamics of the system. This confirms that LGNN can generalize to larger unseen system sizes when trained on a significantly smaller system size. Note that the plots for CLNN are not shown for 8 and 16-links as the architecture cannot exhibit generalizability to larger system sizes. Finally, to push the limits, we infer the dynamics of a 100-link chain (see Fig. 15). We observe that the LGNN trained on 4-link can scale to a 100-link chain with comparable errors, confirming its ability to model large-scale structures. The trajectories of actual and trained models for some of these systems are provided as videos in the supplementary material (see Appendix A.3 for details).

**Generalizability to systems with different edge properties and external drag.** Although the framework presented here is generic, the results were limited to systems with similar edge properties. Further, dissipative forces such as drag were not considered in these systems. In order to evaluate the model to incorporate these effect, we consider a 4-link system with different edge properties (see Appendix A.7)and also a system with drag. We observe that the LGNN presented can model systems with varying link properties and drag with comparable errors (see Figures 8 and 14). These results confirm that the LGNN framework can be used for realistic systems with arbitrary link properties and external dissipative forces.

## 4.3 Zero-shot generalizability

In the conventional LNNs employing feed forward MLPs, the training and test system have the same number of particles and degrees of freedom. In other words, an LNN trained for an $n$-particle system cannot be used to perform inference on an $m$-particle system. In contrast, we show here that

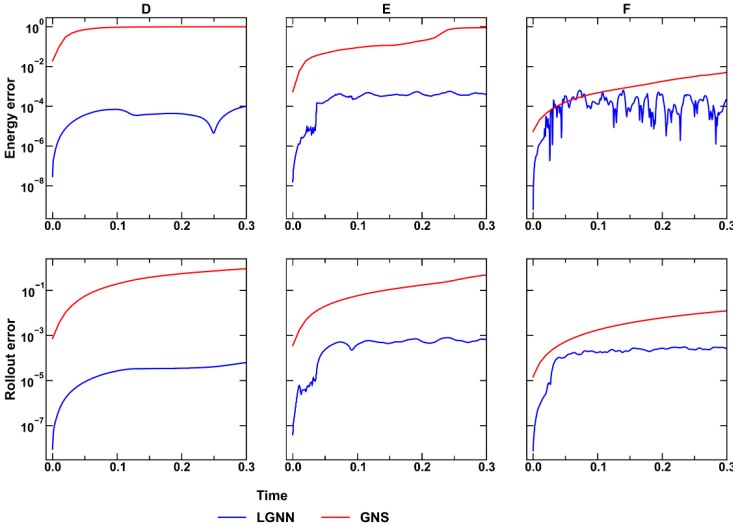

Figure 5: Energy and rollout error of systems predicted using LGNN in comparison with ground truth for T1, T2, and T3. Note that LGNN is trained only on the 4-segment chain and predicted on all the systems. Since the starting configuration of the simulation is fixed (perfect structure as shown in Figure 3), there are no error bars generated for this system.

LGNN trained on a small 4-link system can be used to perform forward simulations on other unseen complex systems such as 100-link system, and tensegrity structures. This ability to infer on different unseen system sizes and topology is referred to as zero-shot generalizability. In order to analyze the zero-shot generalizability of the trained LGNN to simulate complex real-world geometries and structures, we evaluate the ability of LGNN to model the dynamics of tensegrity and lattice-like structures (see Fig. 3). Note that tensegrity structures are truss-like structures comprising of both tension and compression-members. The topology of a tensegrity structure is designed so that the compression members are always bars and the tension members are always ropes. Here, we analyse the ability LGNN to model the equilibrium dynamics of two complex tensegrity structures and the lattice-like structure shown in Figure 3.

To this extent, we use the LGNN trained on the 4-segment structure. We convert the rigid body structure to an equivalent graph and use the trained LGNN to predict the dynamics of the structure when released from the original configuration under gravity. Figure 5 shows the energy error and rollout for both the complex structures and the lattice-like structure shown in Figure 3. We note that the LGNN is able to generalize to a complex structure with varying bar lengths and topology with high accuracy. Specifically, the energy violation and rollout error exhibits very low values for LGNN ($\sim 10^{-4}$). Further, it saturates after a few initial timestep suggesting an equilibrium dynamics. In contrast, we observe that the error in GNS is very high and continues to increase until it reaches 1, which is the maximum it can take. This confirms the superior nature of LGNN to generalize to arbitrary topology, boundary conditions, and bar lengths, after training on a simple 4-segment chain with constant length segments. Visualization of the dynamics of the system T1, predicted by LGNN and the ground truth, is shown in Fig. 6. We observe that the deformed shapes predicted by LGNN are in excellent agreement with the ground truth. Note that since the initial configuration for the forward simulation is fixed, it is not possible to generate error bars for the trajectory.

## 4.4 Nature of the learned mass matrix

Finally, we investigate the nature of the mass matrix of LGNN for different systems. Note that in earlier approaches either the mass matrix was learned directly for a given system based on the EL equations [6], or it was assumed to be diagonal in the Cartesian coordinates [13], or the functional form of kinetic energy was assumed [7]. In the present approach, we do not make any assumptions on the nature of the mass matrix. In fact, for a rigid body, the mass matrix need not be diagonal in nature and depends on the actual topology of the structure. This raises an interesting question about the nature of the mass matrix learned by the LGNN and how it generalizes to arbitrary topologies.

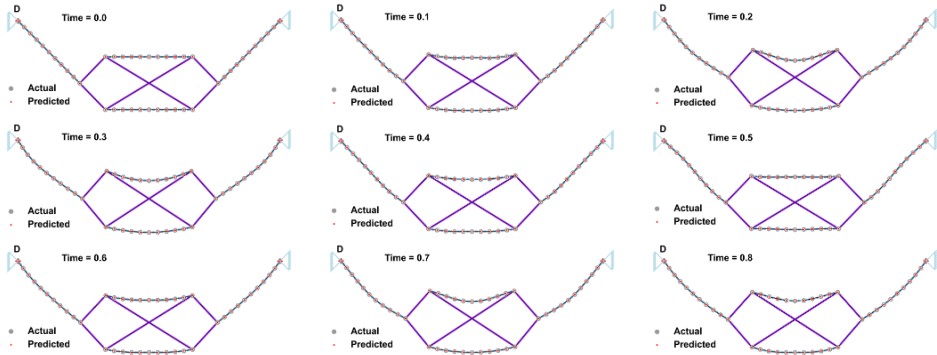

Figure 6: Snapshots of system D during simulation.

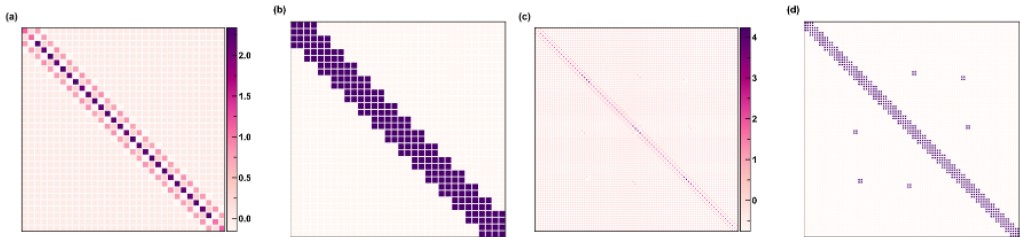

Figure 7: (a) Mass matrix and (b) binary mass matrix showing non-zero values for a chain of 16 nodes. (c) Mass matrix and (d) binary mass matrix showing non-zero values for the complex structure T1 made of chains and rods.

In order to investigate the nature of the mass matrix, we plot the mass matrix of the LGNN in Figure 7. Note that the mass matrix is computed directly from the Lagrangian as $M = \partial^2 \mathcal{L}/\partial \dot{q}^2$, where $\mathcal{L}$ is obtained from the LGNN. First, we analyze the mass matrix of the 16-segment structure. We observe that the mass matrix is banded with a penta-diagonal band as expected for a chain structure. Now, we analyze the mass matrix for a complex structure T1. Interestingly, we observe that the mass matrix learned is non-diagonal in nature and is congruent with the complex topology of the structure (see Figure 7). This confirms that the mass matrix of LGNN is learned on-the-fly during the forward simulation that provides the versatility for LGNN to simulation complex structures.

## 5   Conclusions

In this work, we present a LGNN-based framework that can be used to simulate the dynamics of articulated rigid bodies. Specifically, we present the graph architecture, which allows the decoupling of kinetic and potential energies, that can be used to compute the Lagrangian of the system, which when applied with EL equations can infer the dynamics. We show that LGNN can learn the dynamics from a small 4-segment chain and then generalize to larger system sizes. We also demonstrate the zero-shot generalizability of LGNN to arbitrary topology including a tensegrity structures. Interestingly, we show that LGNN can provide insights into the learned mass matrix, which can exhibit non-trivial structures in complex systems. This suggests the ability of LGNN to learn and infer the dynamics of complex real-life structures directly from the observables such as their trajectory.

**Limitations and future works.** From the mechanics perspective, the LGNN assumes the knowledge of constraints. Learning constraints directly from the trajectory is useful. Similarly, extending LGNN to model contacts, collisions, and deformations allows more comprehesive learning of realistic systems. From the modeling perspective, in our message passing LGNN, all messages are provided equal important. Attention heads in message-passing neural networks have been shown to improve performance remarkably in several domains [28]. We plan to study the impact of attention in LGNN in our future works.

## Acknowledgments and Disclosure of Funding

The authors thank the IIT Delhi HPC facility for providing the computational and storage resources.

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
