# OpenReview forum: "Learning Articulated Rigid Body Dynamics with Lagrangian Graph Neural Network"
_NeurIPS.cc/2022/Conference — NeurIPS 2022 Accept_

### Official Review · Reviewer_EADF · 2022-06-30

**Rating:** 5
**Confidence:** 4
**Soundness:** 2 fair
**Presentation:** 3 good
**Contribution:** 2 fair

**Summary:**

The paper presents a GNN-based Lagrangian formulation for learning the dynamics of chains and ropes, and achieve better accuracy and energy behavior than an unconstrained model (GNS).

**Questions:**

- the terms gamma, a(q) etc. need to be properly defined in the methods section
- why do you even need a full GNN with L layers of message passing? kinetic/potential energy are local, so it at most would need 1 round of collecting neighborhood information
- it sounds like the paper makes a distinction between bars, chains and ropes. In a rigid-body approximation, aren't those exactly the same (the only difference being that ropes having shorter segment lengths than bars) ?
- in the appendix, why is LGN (which I think refers to a _Hamiltonian_ GNN) much worse? The core method is very similar, and I'd expect it to preserve energy very well.

**Limitations:**

The method is quite limited in what it can be applied to, which is pointed out briefly in the conclusion section.

**Strengths And Weaknesses:**

Most papers on HNN, LNN approaches only demonstrate results on mass/spring and n-body system, and the premise of using these methods on more complex, realistic systems is alluring. However, while the paper is motivated by the challenges of rigid-body simulation, I'd argue the method doesn't actually tackle any issue that make rigid-body simulation a hard problem, such as collision between complex object shapes, frictional contact, or even 3D rotation for objects with nontrivial inertial tensor.

Contrary to what the introduction states, 2D ropes can be solved pretty well with simple particle-based methods (see e.g. the 2016 Interaction Nets paper). The non-diagonal mass matrix and inertial tensor is (a) only a problem for method which have an explicit formulation for it (e.g. LNN/HNN), end-to-end learned models (such as plain GNNs) can learn the effect from data and neighborhood relations and (b) the effect disappears in the limit of small edge lengths.

The main contribution of this paper hence is demonstrating that graph-based LNN approaches work for bars/chains/ropes, and make use of the energy-stability of those methods. However, it's worth noting that properties such as length preservation, generalization that the paper shows stem from the fact that very little is learned here; all the constraints (segment length constraint, friction, drag etc.) are manually encoded, only inertial dynamics are learned, and those equations are both universal, and quite straightforward to infer from geometry. Normally you'd want a learned method to do the opposite, i.e. learn complicated interactions/friction models that may be hard to measure for real systems, and manually encode the known priors. The only case I could think of where the proposed approach would be useful is for inferring non-trivial mass distributions which may not be visible (say, on a drawbridge made from composite anisotropic material); but those use cases would need to be demonstrated.

So while I do think there's value in expanding the type of systems that can be tackled with LNN/HNN methods, the paper in its current form is overclaiming its contributions (rigid dynamics) and quite limited in what it adds. It could be extended into a much stronger paper by showing how any of the harder aspects of rigid simulation can be tackled, or how the tricky bits (e.g. constraints, friction, ...) can be learned.

---

> ### Author Response · Authors · 2022-08-02
> **Response to Reviewer EADF**
>
> *Most papers on HNN, LNN approaches only demonstrate results on mass/spring and n-body system, and the premise of using these methods on more complex, realistic systems is alluring. However, while the paper is motivated by the challenges of rigid-body simulation, I'd argue the method doesn't actually tackle any issue that make rigid-body simulation a hard problem, such as collision between complex object shapes, frictional contact, or even 3D rotation for objects with nontrivial inertial tensor.*
>
> **Response:** We thank the reviewer for the critical comment. We agree that the problems reviewer mentioned are interesting and challenging. However, to the best of our knowledge, there are no works where the Lagrangian of a rigid body is learned directly from the trajectory of a very small system and then extrapolated to much more complex systems. To demonstrate this, we performed additional experiments, namely, 100 link chain (see Fig. 13, Appendix A.8), 4-link chain with varying masses and moment of inertia (see Fig. 7, Appendix A.8), and a lattice-like structure (see Fig. 5(f)). The results of these experiments clearly demonstrate that the approaches can be extended to more complex and realistic scenarios. Note that the inference of the dynamics of the 100-link chain and lattice-like structures are performed directly using the model that was trained on a simple 4-link chain. Hence, the present work shows that the LGNN can be used to model real-world complex structures. Further, to address the reviewer's we also performed additional simulation of a 4-link chain with drag (see Fig. 12, Appendix A.8). Results suggest that LGNN can learn the dynamics of a system with drag force on the structure. This suggests that LGNN can be directly extended to some of the complex scenarios suggested by the reviewer. The other problems including collisions and contact will be explored as part of future work.
>
> *Contrary to what the introduction states, 2D ropes can be solved pretty well with simple particle-based methods (see e.g. the 2016 Interaction Nets paper). The non-diagonal mass matrix and inertial tensor is (a) only a problem for method which have an explicit formulation for it (e.g. LNN/HNN), end-to-end learned models (such as plain GNNs) can learn the effect from data and neighborhood relations and (b) the effect disappears in the limit of small edge lengths.*
>
> **Response:** While we agree with the reviewer to a certain extent, the approaches suggested by the reviewer have some flaws as the reviewer himself/herself might be aware. (a) While plain GNNs can learn the effect from data and neighborhood, they do not exhibit a stable long-term trajectory. This is evident from the baseline GNS used in the present study, which shows NaN when extended to a longer duration. In contrast, LGNN exhibits a stable long-term trajectory. In addition, plain GNNs can lead to unrealistic trajectories which do not respect fundamental laws of physics such as energy and momentum conservation, rendering their inferences unreliable. (b) Indeed, the effect disappears in the limit of small edge lengths. However, this limit is applicable only to 2D ropes. In systems such as bars, trusses, chains or tensegrity structures, the edge length can be finite and large. In the present work, we show that the dynamics of articulated rigid bodies can be learned using the LGNN. Thus, the main focus is not just ropes, but also systems having finite lengths. Thus, while we agree with the argument presented by reviewer in the specific case of ropes, it is not reasonable to extend the same argument for other systems demonstrated in the manuscript.

---

> > ### Author Response · Authors · 2022-08-02
> > **part 2**
> >
> > *The main contribution of this paper hence is demonstrating that graph-based LNN approaches work for bars/chains/ropes and make use of the energy-stability of those methods. However, it's worth noting that properties such as length preservation, generalization that the paper shows stem from the fact that very little is learned here; all the constraints (segment length constraint, friction, drag etc.) are manually encoded, only inertial dynamics are learned, and those equations are both universal, and quite straightforward to infer from geometry. Normally you'd want a learned method to do the opposite, i.e. learn complicated interactions/friction models that may be hard to measure for real systems, and manually encode the known priors. The only case I could think of where the proposed approach would be useful is for inferring non-trivial mass distributions which may not be visible (say, on a drawbridge made from composite anisotropic material); but those use cases would need to be demonstrated. So, while I do think there's value in expanding the type of systems that can be tackled with LNN/HNN methods, the paper in its current form is overclaiming its contributions (rigid dynamics) and quite limited in what it adds. It could be extended into a much stronger paper by showing how any of the harder aspects of rigid simulation can be tackled, or how the tricky bits (e.g. constraints, friction, ...) can be learned.*
> >
> > **Response:** As mentioned earlier, in order to demonstrate the performance of LGNN, we have now performed four additional experiments, namely, 100 link chain, 4-link chain with varying masses and moment of inertia, a lattice-like structure, and a 4-link chain with drag. Results suggest that LGNN can be used to learn the dynamics of non-trivial systems.
> > *the terms gamma, a(q) etc. need to be properly defined in the methods section*
> >
> > **Response:** We thank the reviewer for the careful reading. Gamma, A(q), and other terms are now properly defined in the revised manuscript.
> >
> > *why do you even need a full GNN with L layers of message passing? kinetic/potential energy are local, so it at most would need 1 round of collecting neighborhood information*
> >
> > **Response:** We agree with the reviewer. While describing the method in a generic fashion, it was mentioned that the GNN can have L layers of message passing. As mentioned in the Appendix, in the present case, we perform only one round of message passing.
> >
> > *it sounds like the paper makes a distinction between bars, chains and ropes. In a rigid-body approximation, aren't those exactly the same (the only difference being that ropes having shorter segment lengths than bars)?*
> >
> > **Response:** We agree with the comment. Bars, chains, and ropes are different only in the length of the segment. These terms were used to suggest the applicability of the method to different kinds of systems. This is now clarified in the manuscript.
> >
> > *In the appendix, why is LGN (which I think refers to a Hamiltonian GNN) much worse? The core method is very similar, and I'd expect it to preserve energy very well.*
> >
> > **Response:** LGN refers to a baseline which employs exactly the same equations of motion as LGNN but with a full graph network (see: Battaglia, P.W., Hamrick, J.B., Bapst, V., Sanchez-Gonzalez, A., Zambaldi, V., Malinowski, M., Tacchetti, A., Raposo, D., Santoro, A., Faulkner, R. and Gulcehre, C., 2018. Relational inductive biases, deep learning, and graph networks. arXiv preprint arXiv:1806.01261.). The aim of demonstrating the LGN baseline was to study the effect of the specific graph architecture proposed in the present work. With all the training and inference procedure remaining the same, LGN uses a full graph architecture. The results demonstrate the LGNN performs better than LGN suggesting that the superior performance of LGNN is not just due to the inductive biases in the form of EL equations but also due to the specific graph architecture employed.

---

> > > ### Comment · Reviewer_EADF · 2022-08-05
> > > **Updated review**
> > >
> > > Thank you for the response. The additional experiments and comparisons definitely help strengthen the paper.
> > > Although I suspect the new GNS comparison might be an undertuned baseline (maybe trained without noise?), while I expect more energy violation than a Lagrangian method, it shouldn't go unstable on this type of dynamics when trained properly.
> > >
> > > I raised my score slightly to 5, but I remain on the fence with this paper, as I still feel it's advertising itself as something it is not. E.g. the most important properties it shows aren't learned, it's not solving "rigid dynamics" (at the very least, title and abstract should be changed to something ropes/chains), the graph net doesn't actually perform message passing etc.

---

> > > > ### Author Response · Authors · 2022-08-06
> > > > **Thank you for the post-rebuttal feedback**
> > > >
> > > > We thank the reviewer for taking the time to review the response and for the positive comments.
> > > > 1. Indeed, we agree with the reviewer that the work focuses on articulated rigid bodies instead of general bodies. As clarified to Reviewer VUDP, at this stage, we are unable to change the title of the manuscript. We will do this if it is possible. In addition, in the contributions mentioned in the introduction section, we have clarified that the work focuses on articulated rigid body dynamics. In addition, we will also modify the abstract as well. We also request feedback from the reviewer if they have any suggestions for additional changes in this regard. We will be glad to incorporate those.
> > > > 2. Both Lagrangian and GNS are trained on the same dataset. However, GNS might be unstable because the trajectory on which it is trained is much smaller than the rollout trajectory. As such, when tested on a long trajectory, the errors accumulate and eventually explode. This also shows the superiority of LGNN to infer long-timescale behaviour when trained on a smaller trajectory. This will be clarified in the updated manuscript.
> > > >
> > > > Thank you for your time again!
> > > >
> > > > Best regards,
> > > > Authors

---

### Official Review · Reviewer_VUdP · 2022-07-10

**Rating:** 4
**Confidence:** 3
**Soundness:** 3 good
**Presentation:** 3 good
**Contribution:** 2 fair

**Summary:**

In this paper, the authors propose a Lagrangian graph neural network (LGNN) that can learn the dynamics of rigid bodies which can be modelled as a graph. The LGNN consists of two GNNs, which learn the potential energy and kinetic energy of the system Lagrangian respectively. The performance of the model is verified on 2D articulate body simulations based on in-extensible chains and rods. The experiments show that the proposed method can outperform baseline method Graph Neural Simulator (GNS). The generalizability of the model is evaluated on system with larger sizes and unseen topology.

**Questions:**

- The simulations shown can be solved by traditional numerical methods fairly accurately and effectively. I think experiments on larger scale or more complex scenarios where the traditional numerical methods are slow or struggling, will greatly enhance the statements made in the paper.
- The term `rigid body dynamics` is somehow too general I think. Actually the simulations learned here are limited to articulate-body dynamics which can be modelled as a graph.
- The term `zero-shot generalizability` is mentioned in the paper multiple times but without explanations or references. It will be helpful to explain what the `zero-shot` here indicates.
- Videos shown in the supplemental materials demonstrate a simulation of 1s physical time. However, in the paper, the Figure 5 and Figure 6 only show the errors  for 0.1s and 0.3s physical time. It will be more convincing to show how the errors evolving during the whole 1s simulation.
- There are no visual results of the simulation shown in the paper. Putting some key snapshots of the videos will help the readers to easily understand the problems to solve here.
- In the appendix A.6 training details, the author mentioned the baseline method Lagrangian Graph Network (LGN), I am wondering what is the difference between the LGN and proposed LGNN?

Minors:
- Mixed usages of `colon` and `period`, e.g., line 221 `loss function` uses colon while other paragraph titles in the same section uses period; the paragraph titles 5.1 use colon but that of 5.2 use period.



**Limitations:**

The authors have discussed the limitations in the conclusion to some extent.

**Strengths And Weaknesses:**

Strengths
- Learning the Lagrangian mechanics using GNN is interesting and the experiments results and the videos shown in the supplemental materials are neat.
- The paper is written clearly and well organized.
- The introduction to the background knowledge helps understand the proposed idea.

Weakness
- The method is validated on a simple (less than 20 links) rigid body simulation scenario, where the traditional numerical methods can fairly accurately and effectively solve the problem. Therefore, the key point I am concerning is what the advantage of the learning based method proposed comparing to traditional method. The advantages can be either more efficient than traditional simulations or exploring dynamics the traditional simulations do not easy to capture. However, the comparisons on such cases are missing which makes the contribution somehow hard to evaluate. I think experiments on more large scale or complex scenarios will make the proposed method more convincing.

---

> ### Author Response · Authors · 2022-08-02
> **Response to Reviewer VUdP**
>
> *The method is validated on a simple (less than 20 links) rigid body simulation scenario, where the traditional numerical methods can fairly accurately and effectively solve the problem. Therefore, the key point I am concerning is what the advantage of the learning based method proposed comparing to traditional method. The advantages can be either more efficient than traditional simulations or exploring dynamics the traditional simulations do not easy to capture. However, the comparisons on such cases are missing which makes the contribution somehow hard to evaluate. I think experiments on more large scale or complex scenarios will make the proposed method more convincing. The simulations shown can be solved by traditional numerical methods fairly accurately and effectively. I think experiments on larger scale or more complex scenarios where the traditional numerical methods are slow or struggling, will greatly enhance the statements made in the paper.*
>
> **Response:** Traditional approach requires the knowledge of the functions to compute the potential and kinetic energies a priori to learn the dynamics of the system. The main advantage of the present approach is that without the knowledge of any functional forms governing kinetic and potential energy, the dynamics can be learned directly from the trajectory of a small system. This can then be extrapolated to large and complex systems without requiring any additional training.
>
> In order to address the comment regarding *simple systems*, we now perform three additional challenging tasks, namely:
> * inferring the dynamics of 100 link chain from an LGNN trained only on 4-link model (see Fig. 13, Appendix A.8),
> * 4-link chain with varying masses and moment of inertia (see Appendix A.8, Figure 7)
> * 4-link chain with drag (see Appendix A.8, Figure 12)
> * a lattice-like structure (see Figure 5(f)).
> * The results of these experiments clearly demonstrate that the approaches can be extended to more complex and realistic scenarios.
>
> *The term rigid body dynamics is somehow too general I think. Actually the simulations learned here are limited to articulate-body dynamics which can be modelled as a graph.*
>
> **Response:** We agree with the reviewer. However, at this stage we are unable to change the title of the manuscript. To address this comment, in the contributions mentioned in the introduction section we have clarified that the work focuses on articulated rigid body dynamics.
>
> *The term zero-shot generalizability is mentioned in the paper multiple times but without explanations or references. It will be helpful to explain what the zero-shot here indicates.*
>
> **Response:** In the conventional LNNs employing feed forward MLPs, the training and test system have the same number of particles and degrees of freedom. In other words, an LNN trained for an n-particle system cannot be used to perform inference on an m-particle system, which makes these approaches "transductive" in nature. In contrast, we show here that LGNN trained on a small 4-link system can be used to perform forward simulations of other complex systems such as 100-link system, and tensegrity structures. This ability to infer on different *unseen* system sizes and topology is referred to as zero-shot generalizability of LGNN. This has now been clarified in Sec 5.3.
>
> *Videos shown in the supplemental materials demonstrate a simulation of 1s physical time. However, in the paper, the Figure 5 and Figure 6 only show the errors for 0.1s and 0.3s physical time. It will be more convincing to show how the errors evolving during the whole 1s simulation.*
>
> **Response:** The videos in the supplementary material is that of LGNN. The figures in the main manuscript were restricted to shorter duration as the baselines started exploding leading to NaN values. To address this comment, new figures (see Fig.9 and Fig.10) have been added to Appendix A.8, which shows the longer duration simulation. We observe that LGNN provides a stable trajectory with stable error in contrast to GNS which explodes after some time.
>
> *There are no visual results of the simulation shown in the paper. Putting some key snapshots of the videos will help the readers to easily understand the problems to solve here.*
>
> **Response:** Thank you, we have now added some visual snapshots of the simulations in Appendix A.9, Figures 15, 16, and 17.

---

> > ### Author Response · Authors · 2022-08-02
> > **part 2**
> >
> > *In appendix A.6 training details, the author mentioned the baseline method Lagrangian Graph Network (LGN), I am wondering what is the difference between the LGN and proposed LGNN?*
> >
> > **Response:** The aim of demonstrating the LGN baseline was to study the effect of the specific graph architecture proposed in the present work. With all the training and inference procedure remaining the same, LGN uses a full graph architecture (see: Battaglia, P.W., Hamrick, J.B., Bapst, V., Sanchez-Gonzalez, A., Zambaldi, V., Malinowski, M., Tacchetti, A., Raposo, D., Santoro, A., Faulkner, R. and Gulcehre, C., 2018. Relational inductive biases, deep learning, and graph networks. arXiv preprint arXiv:1806.01261.), which has been used to model simple spring and pendulum systems using Hamiltonian graph neural network approaches. The results demonstrate that LGNN performs better than LGN, which suggests that the superior performance of LGNN is not just due to the inductive biases in the form of EL equations but also due to the specific graph architecture employed.
> >
> > *Mixed usages of colon and period, e.g., line 221 loss function uses colon while other paragraph titles in the same section uses period; the paragraph titles 5.1 use colon but that of 5.2 use period.*
> >
> > **Response:** We thank the reviewer for the careful reading. All the formatting errors have now been corrected.

---

> ### Author Response · Authors · 2022-08-06
> **Looking forward to your post-rebuttal feedback**
>
> Thank you once again for your insightful suggestions and comments. As the deadline for discussion is approaching, we are glad to provide any additional clarifications that you may need.
>
> In our previous response, we have carefully studied your comments and added a lot more experiments and analyses to complement your suggestions. We summarize the major additional experiments performed to address the reviewer's concerns below:
>
> 1. We demonstrate how LGNN trained on a simple system can be extended to a system that is **~25 times larger**. Specifically, we infer the **dynamics of 100 link chain** from an LGNN trained only on the 4-link model (see Fig. 13, Appendix A.8),
> 2. We show that LGNN can infer **the dynamics of a complex lattice-like structure** after training on a simple 4-link model (see Figure 5).
> 3. We learn the dynamics of **4-link chain with varying lengths, masses, and moment of inertia** (see Appendix A.8, Figure 7)
> 4. We show that LGNN can learn the dynamics of **4-link chain with drag** (see Appendix A.8, Figure 12)
> 5. We have **extended the figures** for the entire **1s duration** to show the error evolution clearly.
> 6. **Visual plots** of all the complex simulations are now included in the Appendix.
>
> In addition, we have also mentioned in the manuscript that the work focuses on articulated rigid body mechanics as opposed to general rigid body mechanics. We hope that the provided new experiments and additional explanations have convinced you of the merits of our work. Please do not hesitate to contact us if there are other clarifications or experiments we can offer.
>
> Thank you for your time again!

---

> > ### Comment · Reviewer_VUdP · 2022-08-07
> > **Updated review**
> >
> > Thanks for the detailed response. The additional experiments provided are  appreciated. The explanation for the zero-shot generalizability also improve the presentation of the paper.
> >
> > As the problem studied i.e., simple 2D articulate body dynamics, is a problem well solved by traditional numerical methods, it is important that the proposed method should show some advantages from either accuracy or efficiency. However, the proposed method seems not to outperform traditional methods from both sides. Though the 100 links chain example is indeed more complex than the original ones in the paper, it is still too much easier comparing to simple real world 2D cases such as 10k DoF, which is not convincing.
> >
> > Due to the reasons mentioned above, I will keep my score unchanged.

---

> > > ### Author Response · Authors · 2022-08-09
> > > **Thank you for the response**
> > >
> > > We thank you for raising the additional concern regarding more complex systems. We would like to highlight that the main contribution of the present work is to **learn the dynamics** of articulated rigid bodies **directly from their trajectory**. Traditional approaches such as the ones mentioned by the reviewer require a priori knowledge of the functional forms of the Lagrangian/Hamiltonian. Here, we show that the Lagrangian is learned during the training. Hence, the advantage of the approach is not necessarily in terms of accuracy or efficiency, but in the ability to **learn abstract quantities such as Lagrangian or Hamiltonian from trajectory**. Even in the case of complex systems with varying masses and moments of inertia or subjected to dissipative drag forces, LGNN is able to learn the dynamics. More interestingly, we show that once learned on a small system such as a 4-link chain, it can extrapolate it to **more complex structures** and **infer non-trivial mass matrices**. Indeed, in the specific case, we extended it to 100 link chains as an example that is 25 times larger than the original system. It is indeed possible to extend it to a 10k DoF system as well. However, due to the unavailability of time, we were unable to complete the experiments during the author-reviewer discussion period. This can be included in the final version of the manuscript.
> > >
> > > We thank the reviewer for engaging in a fruitful discussion and for the comments, which have significantly improved the quality of the manuscript. Please do let us know if there are any further clarifications that we can offer. Thank you.

---

### Official Review · Reviewer_6uQz · 2022-07-11

**Rating:** 8
**Confidence:** 4
**Soundness:** 3 good
**Presentation:** 3 good
**Contribution:** 4 excellent

**Summary:**

A novel framework is introduced for Lagrangian Neural Networks using graph networks to model rigid body dynamics. Comparisons are made with classical Lagrangian and Hamiltonian Neural Networks, showing the generalizability to arbitrary topologies and system sizes.

**Questions:**

1) Same from before: line 100 the $\nabla_{\dot{q}_i \dot{q}_i}$ operator, would that be a second-order derivative?
2) How did the choice of Section 4.1 come about, where the edges represent rigid bodies and the nodes represent connections between the rigid bodies? What benefit/drawback does this have?
3) Did you perform any experiments on the Hamiltonian extensions?
4) Figure 5 shows a quasi constant error between the two methods, is there a reason for this? Is this not the initial error propagating throughout the rollout trajectory?
5) Why is the GNS result in Figure 6 so smooth, while the LGNN is more chaotic?

**Limitations:**

The limitations are clearly addressed, though focusing more on that contact modeling is a limitation would be preferred, since the main focus is rigid body mechanics, where contact and collision is essential. Adding some initial ideas towards addressing contact would be appreciated as well.

**Strengths And Weaknesses:**

The contributions are very clear, and the authors did a great job explaining their framework for rigid bodies. The benchmarks are clear, and the non-diagonal mass matrices show the capabilities nicely of the model. Some steps could, however, involve a few more explanations for readers that are not familiar with the topic, such as in line 100 the $\nabla_{\dot{q}_i \dot{q}_i}$ operator, would that be a second-order derivative? It is not a notation I have seen before. Also, in line 121 you mention solving and substituting the lambda, it would be great if this derivation were present in the Appendix.

The information presented in Figure 2 and 3 were a bit overlapping, and it could perhaps be combined to make the paper visuals more concise. Figure 4 also has text "ABC" which are not really explained (can be inferred from caption), and ideally the purple lines can be explained in the caption as well.

The extension to the Hamiltonian feels oversimplified, though I'm not sure about the topic. It would absolutely be worth investigating. If no experiments were performed, section 4.2 could be left out.

In line 276 the authors refer to an appendix, but no appendix was present in the file.

All in all, more results with exciting articulated bodies could strengthen the paper a lot, and especially scenarios such as Pfaff et al. 2020 (Learning mesh-based simulation with Graph Networks) with deforming plate and waving flag can be interesting to try.

---

> ### Author Response · Authors · 2022-08-02
> **Response to Reviewer 6uQz**
>
> *The contributions are very clear, and the authors did a great job explaining their framework for rigid bodies. The benchmarks are clear, and the non-diagonal mass matrices show the capabilities nicely of the model. Some steps could, however, involve a few more explanations for readers that are not familiar with the topic, such as in line 100 the ∇q˙iq˙i operator, would that be a second-order derivative? It is not a notation I have seen before. Also, in line 121 you mention solving and substituting the lambda, it would be great if this derivation were present in the Appendix.*
>
> **Response:** Thank you for the positive comments on the manuscript. Indeed,∇q˙iq˙i operator represents a second derivative. To make the notation clear, these operators are now written using the standard notation employing $\partial$. The text in the manuscript is updated accordingly. Further, the derivation for obtaining lambda is now included in Appendix A.3.
>
> *The information presented in Figure 2 and 3 were a bit overlapping, and it could perhaps be combined to make the paper visuals more concise. Figure 4 also has text "ABC" which are not really explained (can be inferred from caption), and ideally the purple lines can be explained in the caption as well.*
>
> **Response:** We have now combined Figures 2 and 3. In addition, all the subfigures are properly labeled and explained in the caption as well.
>
> *The extension to the Hamiltonian feels oversimplified, though I'm not sure about the topic. It would absolutely be worth investigating. If no experiments were performed, section 4.2 could be left out.*
>
> **Response:** We agree with the reviewer that no experiments on extension to Hamiltonian is carried out as it would increase the scope of the manuscript significantly. As such, this extension is kept as part of future work. As of now, section 4.2 is now removed from the manuscript.
>
> *In line 276 the authors refer to an appendix, but no appendix was present in the file.*
>
> **Response:** Appendix was provided as a separate file in the supplementary material. In the updated manuscript, the Appendix is also included in the same file as the main manuscript.
>
> *Same from before: line 100 the ∇q˙iq˙i operator, would that be a second-order derivative?*
>
> **Response:** Yes. This notation has now been removed from the main manuscript.
>
> *How did the choice of Section 4.1 come about, where the edges represent rigid bodies and the nodes represent connections between the rigid bodies? What benefit/drawback does this have?*
>
> **Response:** The main advantage of a graph architecture is its inductivity to larger system sizes. In articulate rigid body, small substructures, such as bars, spheres or other such elements are connected to each other through hinges or pins. These pins allow the degrees of freedom in these rigid, just like joints in robotic systems. Note that in articulate systems, a single joint can be connected to many elements, but one element is connected only to two joints. For instance, in a chain or a truss, a link is connected only to two joints, but several links can be connected to one joint. Intuitively, we propose a graph architecture, where the nodes represent these joints, that govern the degrees of freedom of the system, and edges represent the rigid body. Note that this approach has a drawback in cases where a elements represented by edges are connected to more than two joints, for instance, a triangular element. In such cases, we may need to reformulate the graph architecture to accommodate these joints.
>
> *Did you perform any experiments on the Hamiltonian extensions?*
>
> **Response:** No. In the present manuscript, no experiments are performed on the extension to the Hamiltonian. Hence the section 4.2 is removed from the manuscript. This will be pursued as part of a future study.
>
> *Figure 5 shows a quasi-constant error between the two methods, is there a reason for this? Is this not the initial error propagating throughout the rollout trajectory?*
>
> **Response:** The initial error between the two methods is zero as they start from exactly the same configuration. Note that the y-axis of Figure 5 is in log scale. Hence, although the initial error is zero, it is not visible. Further, the error between the two methods increases by orders of magnitude as the difference between the errors should also be read in log-scale. To clarify this point, we have now plotted the absolute error in addition to the relative error in the Appendix A.8. in Figs. 8 and 9, which clearly shows that the error between the two methods is not constant.

---

> > ### Author Response · Authors · 2022-08-02
> > **part 2**
> >
> > *Why is the GNS result in Figure 6 so smooth, while the LGNN is more chaotic?*
> >
> > **Response:** GNS is poor in predicting the trajectory and hence hits the upper bound of the relative error, that is 1, quickly. On the other hand, in the case of LGNN, the model can capture the chaotic behavior with low error in both energy and rollout. The fluctuations in the case of LGNN is due to the fact that the simulation is performed only for a given configuration. It is not possible to perform multiple forward simulations as we are simulating the dynamics of the system when it is released from the perfect structure conditions.

---

> > > ### Comment · Reviewer_6uQz · 2022-08-07
> > > **Thank You for the Revision**
> > >
> > > Thank you for your response and taking the time to adjust the manuscript accordingly! Be careful with the new figure 2 though, since the text tends to be quite small, and it should remain readable. After the removal of section 4.2, it might be worth considering not having a "4.1" and just add it all to section 4. The revision is very much appreciated!

---

> > > > ### Author Response · Authors · 2022-08-08
> > > > **Thank you**
> > > >
> > > > Thank you for the careful reading and the positive comments, we really appreciate it. We have now corrected the figure, and removed "4.1" to combine it with section 4. The submission has now been updated with the latest version of the manuscript.

---

### Official Review · Reviewer_aiXF · 2022-07-12

**Rating:** 6
**Confidence:** 4
**Soundness:** 3 good
**Presentation:** 3 good
**Contribution:** 2 fair

**Summary:**

The authors introduce a Lagrangian graph neural network to predict the trajectory of a rigid body. The rigid body is represented as a graph where edges represent rigid bodies – e.g., links in a chain – and nodes represent connections between those rigid bodies. The position and velocity of each node is assumed to be given. From the input graph, the authors construct two graph neural networks which are trained simultaneously. One learns to predict the kinetic energy of the rigid body and the other learns to predict the potential energy of the rigid body and the Lagrangian is formed by computing the difference of these quantities. The networks are trained by minimizing the difference between the predicted acceleration (determined by the EL equations) and the ground truth acceleration of the system. Against previous work, the proposed approach shows strong predictive performance and an ability to generalizability to unseen structures. The mass and moment of inertia of all members in the graph are the same irrespective of their length.

**Questions:**

- Questions:
  - [55] should it be \psi = sin^(-1)((y_i - y_{i-1})/x_i - x_{i-1}))?
  - Eqn 5: it looks like the kinetic energy is mass times position, shouldn’t it be mass times velocity
  - [208] what is || in eqn 9?
  - [309] How does the learned mass matrix compare to the gt mass matrix?
- Comments:
  - [47] Possibly relevant citations [1].
  - The notation in eq 5 is a bit challenging (e.g., x_i^cm2)
  - Consider using \mathcal{l}(\ddot{q}, \hat{\ddot{q}}) for the loss since \mathcal{L} is used for Lagrangian
- Possible typos:
  - [37] abbreviation EL is used before it is defined
  - [143] “This is because…”
  - [152] “significantly smaller that” → “significantly smaller than”
  - [184, 185] node → nodes
  - [211] N_i = {u_j | (u_i, u_j) \in E}
  - [215] “denoted as” → “denoted by”

[1] Duong, Thai, and Nikolay Atanasov. "Hamiltonian-based Neural ODE Networks on the SE (3) Manifold For Dynamics Learning and Control." Robotics: Science and Systems (RSS). 2021.


**Limitations:**

The authors note that the proposed approach assumes knowledge of system constraints.

**Strengths And Weaknesses:**

- Originality: The method appears to be new. The related work is well organized and the proposed approach is adequately situated in the context of existing methods.
- Quality: The submission appears to be technically sound, with the claims well supported by the empirical analysis.
- Clarity: The paper is well written and organized
- Significance: The paper addresses the issue of predicting physically plausible trajectories of a rigid body system, a challenging problem which has the potential to impact scientific discovery. However, there are several limitations of the approach, i.e., The mass and moment of inertia of all members in the graph are the same, and the constraints of the system are assumed to be known.

---

> ### Author Response · Authors · 2022-08-02
> **Response to Reviewer aiXF**
>
> *Significance: The paper addresses the issue of predicting physically plausible trajectories of a rigid body system, a challenging problem which has the potential to impact scientific discovery. However, there are several limitations of the approach, i.e., The mass and moment of inertia of all members in the graph are the same, and the constraints of the system are assumed to be known.*
>
> **Response:** We thank the reviewer for the positive comments and feedback. To address the issue regarding the mass and moment of inertia being same for all the members, we have now extended the LGNN framework to learn members with varied masses and moment of inertia. Specifically, we train on a 4-link system with each link having different mass, moment of inertia, area of cross section, and length. We show that the LGNN framework can learn the dynamics of the system (see Appendix A.8 and Figure 12).
>
> *[155] should it be $\psi = sin^{-1}((y_i - y_{i-1})/x_i - x_{i-1}))$?*
>
> **Response:** We thank the reviewer for pointing this out. The text has now been corrected as $\psi = tan^{-1}((y_i - y_{i-1})/x_i - x_{i-1}))$.
>
> *Eqn 5: it looks like the kinetic energy is mass times position, shouldn’t it be mass times velocity*
>
> **Response:** Thanks for the careful read. The typo has now been corrected.
>
> *[208] what is $||$ in eqn 9?*
>
> **Response:** $||$ refers to concatenation operation. This has now been clarified in the text.
>
> *[309] How does the learned mass matrix compare to the gt mass matrix?*
>
> **Response:** Thank you for the interesting question. We notice that the learned mass matrix exhibits the same structure (that is, having the same non-zero terms) as the ground truth. In addition, the maximum error in the learned mass matrix for individual is found to be ~1%. Thus, it can be said that the learned mass matrix is very close the ground truth mass matrix. To address this comment, the percentage error of the learned mass matrix is now included as Figure 14 in the Appendix A.8.
>
> *[47] Possibly relevant citations [1]*.
>
> **Response:** Thank you for sharing this relevant citation. It has now been added to the paper.
>
> *The notation in eq 5 is a bit challenging (e.g., x_i^cm2)*
>
> **Response:** Thank you. We have now corrected the notation as $x_{i,cm}^2$. We hope this notation is easier to read and interpret.
>
> *Consider using $\mathcal{l}(\ddot{q}, \hat{\ddot{q}})$ for the loss since $\mathcal{L}$ is used for Lagrangian*
>
> **Response:** Thank you once again for the careful reading. We have now corrected the loss as $\mathbb{L}$.
>
> *Possible typos:*
> *[37] abbreviation EL is used before it is defined*
> *[143] “This is because…”*
> *[152] “significantly smaller that” → “significantly smaller than”*
> *[184, 185] node → nodes*
> *[211] $N_i = {u_j | (u_i, u_j) \in E}$
> *[215] “denoted as” → “denoted by”*
>
> *[1] Duong, Thai, and Nikolay Atanasov. "Hamiltonian-based Neural ODE Networks on the SE (3) Manifold For Dynamics Learning and Control." Robotics: Science and Systems (RSS). 2021.*
>
> **Response:** Thank you for the careful read. All the typos have been corrected and the new reference has been added to the manuscript.

---

> > ### Comment · Reviewer_aiXF · 2022-08-08
> > **Post-rebuttal comments**
> >
> > After reading the authors’ rebuttal and comments from other reviewers I have a the following minor comment:
> > * [Response to Reviewer 6uQz] The old notation for the second derivative is used on [330]

---

> > > ### Author Response · Authors · 2022-08-08
> > > **Thank you.**
> > >
> > > Thank you for the careful reading, which is really appreciated. We have now corrected this equation in the manuscript and updated the submission.

---

### Author Response · Authors · 2022-08-02
**General Comments. Applicable to all reviewers.**

We thank the reviewers for the comments and suggestions. Please find a point-by-point response to the comments raised by the reviewers below. We have also updated the main manuscript and the appendix to address these comments. The changes made in the main manuscript are highlighted in blue color. Some of the major changes are listed below.
1. We demonstrate that the LGNN can learn the dynamics of a chain with **links having varying masses and moment of inertia**. To this extent, we consider a 4-link chain with each link having different mass, length, and moment of inertia and show that LGNN can learn the dynamics.
2. We show that LGNN scales to large system sizes by performing inference on a 100-link chain, a system that is **~25 times larger** than the training system.
3. We show that LGNN can be used to infer the dynamics of **complex lattice-like structure**, by training on a simple 4-link chain.
4. Finally, we show that LGNN can be used in **systems with dissipative forces** such as drag. To this extent, we learn the dynamics of a 4-link chain subjected to linear drag. We show that LGNN can learn the dynamics of this system.

---

### Meta-Review · Area_Chair_p4bs · 2022-08-25

**Recommendation:** Accept
**Confidence:** Less certain

**Metareview:**

The authors propose the use of two separate networks to learn the kinetic and potential energy of objects made of chains of rigid bodies. Their neural net architecture uses knowledge of constraints in the system, and improves on previous GNS work.
Reviewers pointed out that experiments were too simple with a small number of DoFs where traditional methods already perform very well. The authors replied that their method can learn from trajectories only, and include complex settings, for example with dissipative drag forces. They also added a number of much more complex experiments in a revised version.
Overall, this is a borderline paper given the high standard required for NeuIPS publication. I lean towards recommending this paper for acceptance, with one condition: as pointed out by reviewers VUdP and EADF, the title and abstract should be changed to reflect the fact that a) this paper is not solving "rigid dynamics", but only works on "articulated rigid body"; b) the graph net doesn't actually perform message passing.

**Award:**

No

---

### Decision · Program_Chairs · 2022-09-14

Accept